# Ionospheric signatures of a Bursty Bulk Flow in the 6D Vlasiator simulation

Abiyot Workayehu<sup>1</sup>, Minna Palmroth<sup>1,2</sup>, Maxime Grandin<sup>2</sup>, Liisa Juusola<sup>2</sup>, Markku Alho<sup>1</sup>, Ivan Zaitsev<sup>1</sup>, Venla Koikkalainen<sup>1</sup>, Konstantinos Horaites<sup>3</sup>, Yann Pfau-Kempf<sup>4</sup>, Urs Ganse<sup>1</sup>, Markus Battarbee<sup>1</sup>, and Jonas Suni<sup>1</sup>

**Correspondence:** Abiyot Workayehu (abiyot.workayehu@helsinki.fi)

**Abstract.** Bursty Bulk Flows (BBFs) are transient plasma flows in the Earth's magnetotail plasma sheet. These short-lived, high-speed flows play a key role in the magnetosphere-ionosphere coupling. Currently, most insights into the ionospheric signatures of BBFs come from individual case studies that include conjugate observations of BBFs in the magnetotail and fieldaligned currents (FACs) in the nightside ionosphere. In this study, we utilise the 6D hybrid-Vlasov simulations to study the ionospheric signatures of BBFs in the near-Earth magnetotail. We show that a BBF with  $V_x \ge 400 {\rm km/s}$  emerges shortly after magnetic reconnection occurs on the duskside at a radial distance between 11 and 14  $R_{\rm E}$  (where  $R_{\rm E}=6371$  km is the radius of the Earth) in the current sheet. As the BBF moves Earthward, clockwise (counterclockwise) flow vortices are induced on its dawn (dusk) sides. These vortical flows generate FACs flowing upward (out of the current sheet) on the dawnside and downward (into the current sheet) on the duskside flank, respectively. The mapping of BBF structures onto the ionosphere shows that the structure is primarily aligned in the east-west direction, with its ionospheric signatures appearing as enhancements in FACs, ionospheric conductances, horizontal ionospheric currents, energies of precipitating electrons and protons, and the formation of localised plasma flow channels. The upward and downward FACs associated with BBFs in the magnetotail consistently map to enhanced Region 2 (R2) and Region 1 (R1) FAC structures at ionospheric altitude, which are then closed in the ionosphere by north-west flowing Pedersen currents. The ionospheric counterpart of the Earthward plasma flow of the BBF is a channel of equatorward plasma flow, while the westward drift of these enhanced structures corresponds to the duskward motion of the BBF in the magnetotail.

# 1 Introduction

Bursty Bulk Flows (BBFs) are high-speed ( $\geq 400\,\mathrm{km/s}$ ) plasma flows occurring within the Earth's magnetotail plasma sheet (Angelopoulos et al., 1992). They play a significant role in transporting energy, mass, and magnetic flux throughout the magnetotail (Angelopoulos et al., 1994). BBFs are found to be generated during periods of magnetic reconnection in the magnetotail, a process during which the release of stored magnetic energy accelerates plasma flows in both the Earthward and

<sup>&</sup>lt;sup>1</sup>University of Helsinki, Helsinki, Finland

<sup>&</sup>lt;sup>2</sup>Finnish Meteorological Institute, Helsinki, Finland

<sup>&</sup>lt;sup>3</sup>Cooperative Institute for Research in Environmental Sciences, University of Colorado, Boulder, Boulder, CO, USA

<sup>&</sup>lt;sup>4</sup>CSC – IT Center for Science, Espoo, Finland

tailward directions (Baumjohann, 1990). As BBFs propagate Earthward, they encounter a strong dipolar magnetic field, which decelerates them, and this deceleration redistributes energy across the magnetotail and influences the coupled ionosphere. BBFs are also associated with particle precipitation and FACs (Panov et al., 2013; Gabrielse et al., 2014, 2023, references therein), and significantly affect the high-latitude ionosphere. Thus, the identification and analysis of the ionospheric signatures of BBFs has been a key area of research aimed at understanding the coupling mechanisms between the magnetosphere and ionosphere, as well as its implications for space weather phenomena.

Progress has been made in identifying the ionospheric and ground signatures of BBFs using both observational and simulation data (Kauristie et al., 1996, 2000; Amm and Kauristie, 2002; Sergeev, 2004; Juusola et al., 2009; Pitkänen et al., 2011; Juusola et al., 2013; Yu et al., 2017; Forsyth et al., 2020; Ferdousi et al., 2021). In these studies, auroral streamers, FACs, geomagnetic field disturbances, and ionospheric flow channels are among the commonly reported signatures of BBFs. Auroral streamers are narrow, fast-moving auroral structures that propagate equatorward and westward (Sergeev et al., 2004), and are the commonly reported signature of BBFs. These are thought to reflect the Earthward flow of plasma within the magnetotail and its interaction with the ionosphere. Studies have shown that these are clear manifestations of the ionospheric signatures of the BBFs (Forsyth et al., 2020, and references therein).

Using a combination of observational data and empirical magnetospheric models, Sergeev et al. (2020) investigated the origins and orientations of nightside auroral arcs. They found that the majority of nightside arcs originate from the magnetotail current sheet region and argued that magnetospheric flow channels, such as BBFs, are the most likely source of these arcs. Furthermore, they demonstrated that structures, which appear nearly sun-aligned in the plasma sheet, become increasingly azimuthally aligned when mapped to the ionosphere.

Yu et al. (2017) investigated the effect of BBFs on global-scale currents including FACs, ring current and substorm current wedge (SCW) using global magnetohydrodynamic (MHD) simulation coupled with a kinetic ring current model. Their results indicated that BBFs induce vortices at two different radial distances on the equatorial plane: near the edges of the braking region at  $X = -10R_{\rm E}$  and at the inner magnetosphere around  $X = -6R_{\rm E}$ . The authors noted that vortices in both regions are associated with FACs. However, the FACs near the braking region form the large-scale SCW current system, while the FACs generated by the vortices at the inner magnetosphere are shown to produce localised R1-sense pair of FACs as ionospheric signatures of BBFs.

Based on conjugate magnetospheric and ionospheric observational data, as well as MHD simulations, most of the above-mentioned studies have focused on the distinct ionospheric signatures of BBFs. However, the absence of simultaneous magnetosphere-ionosphere observations linking BBFs to their ionospheric manifestations presents challenges in accurately mapping BBFs in the magnetotail to corresponding features in the ionosphere. Furthermore, many simulation studies are restricted to local boxes without a self-consistent magnetosphere-ionosphere configuration (e.g., Wang et al., 2024) or are limited in their physical descriptions, commonly relying on the MHD approximation (e.g., Merkin et al., 2019). Consequently, there remains a gap in self-consistent global kinetic simulations of the magnetosphere coupled with the ionosphere to investigate the ionospheric signatures of BBFs.

In this study, we used a global 6D (3D in ordinary space and 3D in velocity space) hybrid-Vlasov simulation coupled with an electrostatic ionosphere model to investigate the ionospheric signatures of BBFs. To our knowledge, this is the first study to employ a global hybrid-Vlasov model that uses an ion-kinetic description (Ganse et al., 2025) coupled with a description of ionospheric electrodynamics through its inner boundary, where the ionosphere directly influences the magnetosphere simulation domain and vice versa. We mapped BBF structures in the plasma sheet to the ionospheric altitude along magnetic field lines. This mapping process enables us to determine the alignment of the BBF structure at ionospheric altitude and the conjugacy between the BBF and its ionospheric manifestations. The remainder of this paper is organised as follows: In Section 2, we provide descriptions of the Vlasiator and the ionospheric models; in Section 3, we present the simulation results, in Section 4 we present the discussion and finally, summary of the main results and conclusions are presented in Section 5.

#### 65 **2** Model

# 2.1 Vlasiator

Vlasiator is a global hybrid-Vlasov model used to simulate near-Earth space plasmas (Palmroth et al., 2018; Ganse et al., 2023). In Vlasiator, ions are represented by their velocity distribution function that evolves over time in accordance with the Vlasov equation, while electrons are treated as a massless charge-neutralising fluid (Palmroth et al., 2018).

The simulation used in this study is carried out within the physical boundaries  $[-110, 50]R_{\rm E}$  in the x-direction and  $[-58, 58]R_{\rm E}$  in the y- and z-directions in the Geocentric Solar Magnetospheric (GSM) coordinate system. The inner boundary that couples the hybrid-Vlasov domain with an ionospheric model, which will be discussed in more detail in Section 2.2, is located at  $4.7R_{\rm E}$  from the centre of the Earth. The initial background magnetic field is the unscaled unperturbed geomagnetic dipole whose axis is aligned with the Z-axis of the GSM coordinate system, corresponding to a dipole tilt angle of  $0^{\circ}$ . The simulation setup uses steady and homogeneous solar wind conditions: with a solar wind velocity of  $750 \, \rm km/s$  in the -x direction, a purely southward interplanetary magnetic field of  $5 \, \rm nT$ , a proton number density of  $10^6 \, \rm m^{-3}$ , and a solar wind temperature of  $5 \times 10^5 \, \rm K$ . The fast solar wind was chosen to speed up the initialisation phase of the simulation run (Palmroth et al., 2023), while  $\rm B_z = -5 \, nT$  represents conditions favourable for magnetic reconnection without being strongly disturbed.

# 2.2 The Vlasiator Ionosphere

The Vlasiator model has recently been added with an ionospheric model (Ganse et al., 2025), which enables a two-way magnetosphere-ionosphere coupling. In this framework, the ionosphere is modelled as a thin spherical shell, defined by a constant radial distance of  $R_{\rm i} = R_{\rm E} + 100\,\rm km$  from Earth's centre. This is a commonly used approximation of the ionosphere, as noted in the literature (Vanhamäki et al., 2020, and references therein). The validity of this approximation has been justified by the fact that horizontal currents flowing in the ionosphere are concentrated in the region where the Pedersen and Hall conductivities reach their peak values, specifically between 100 and 150 km altitude. Consequently, the effective thickness of

this region is small when compared to the horizontal length scale of typical ionospheric current systems (Vanhamäki et al., 2020).

The main variables of the Vlasiator ionosphere include those that are mapped down from the magnetosphere (FACs, electron number density, and temperature) and variables calculated using the ionospheric solver on the ionospheric grid (e.g., ionospheric horizontal current density, electrostatic potential, Hall and Pedersen conductances) (Ganse et al., 2025). The FAC density is calculated from the simulation magnetic field at  $5.6\,R_{\rm E}$  (i.e.,  $0.9\,R_{\rm E}$  away from the inner boundary) using Ampère's law, while the electron temperature ( $T_{\rm e}$ ) is related to the magnetospheric ion temperature ( $T_{\rm i}$ ) by  $T_{\rm e} = \frac{T_{\rm i}}{4}$ .

In the Vlasiator ionosphere, conductivities are produced by three sources: photoionisation by solar extreme ultraviolet (EUV) radiation, ionisation due to precipitating energetic particles, and galactic cosmic rays. The conductivities due to solar EUV radiation depend on the solar zenith angle and solar radio flux values and are calculated using a model by Moen and Brekke (1993). The conductivity due to particle precipitation is calculated using the parametrised precipitating electron energy flux and ionospheric neutral density profile from the NRLMSISE model (Picone et al., 2002). The contribution of galactic cosmic rays to the ionospheric conductances is taken as a constant  $\Sigma_{\rm H,P} = 0.5\,\rm S\,m^{-1}$ . Details of the calculation of the height-integrated conductivities can be found in Ganse et al. (2025).

In Vlasiator, the magnetosphere and ionosphere are coupled as follows:


- 1. FACs are computed from the simulation magnetic field at  $5.6R_{\rm E}$  from the centre of the Earth.
- 2. FACs, electron number density and electron temperature are mapped along the dipole magnetic field lines to the ionospheric grid at an altitude of  $h=100~\mathrm{km}$ .
- 3. The downmapped variables are provided as input to the ionospheric model (Ganse et al., 2025), which gives the ionospheric conductances, horizontal currents and electrostatic potential  $\Phi$  as output. As the ionosphere is modelled as a thin spherical shell, the height-integrated ionospheric horizontal current J is related to the conductance tensor  $\Sigma$ , the electric field E and electrostatic potential  $\Phi$  by  $J = \Sigma \cdot E = \Sigma \cdot (-\nabla \Phi)$ . Also, from the current continuity condition, the divergence of the ionospheric horizontal current must be closed by the downmapped FAC as  $\nabla \cdot J = \nabla \cdot [\Sigma \cdot (-\nabla \Phi)] = -\text{FAC}$ .
- 4. The electric field  $(-\nabla\Phi)$  is calculated from  $\nabla\cdot[\mathbf{\Sigma}\cdot(\nabla\Phi)]=\mathrm{FAC}$ , and then mapped back to the magnetospheric simulation domain at the inner boundary. This electric field is used to calculate the ion drift velocity, which subsequently modifies the ion velocity distribution functions near the inner boundary.

In addition to the primary variables of the Vlasiator ionosphere mentioned above, we also utilise proton (0.5–50 keV) and electron precipitation fluxes as indicators of the ionospheric signatures of BBFs, as discussed in Section 3.3. The proton precipitation fluxes are calculated within the simulation domain using the method presented in Grandin et al. (2023, references therein). Meanwhile, the energy fluxes of the precipitating electrons are computed within the ionospheric grid, as outlined in Ganse et al. (2025). In this study, the proton fluxes calculated at 5.6  $R_{\rm E}$  are mapped to the ionospheric altitude along the magnetic field lines.

# 3 Simulation Results






# 3.1 General properties of the BBF

We investigate in detail a BBF that initially forms around t=450 s in the simulation used in this study. In this section, we present the general properties of the BBF in the magnetotail from its generation to its subsequent propagation from the tail towards Earth. The flow burst is identified during temporal changes in magnetic topology associated with magnetic reconnection. This process is characterised by the conversion of magnetic energy into the kinetic and thermal energy of the plasma. In this context, we examine the flow velocity, magnetic field, and plasma pressure gradient.

Figure 1 shows a zoomed-in snapshot of velocity, magnetic field and magnitude of pressure gradient in the equatorial plane of the magnetotail, covering the region from -20  $R_E$  to +1  $R_E$  along the x-axis and from -10 $R_E$  to +10 $R_E$  along the y-axis for different simulation times. From left to right, the first, second, and third columns show the values of the x-component of the ion bulk velocity  $V_x$ , the z-component of the magnetic field  $B_z$  that is perpendicular to the equatorial plane, and the magnitude of the plasma pressure gradient in the equatorial plane. In all panels, the black contour lines represent the  $B_z$  values, while the white contour lines indicate  $B_z = 0$  nT. In this section, the  $B_z = 0$  contour line is used as a proxy for suggesting the reconnection line (discussed later in Section 3.2). The magenta contour lines indicate the locations of the fast plasma flow regions in the magnetospheric equatorial plane, where the velocity  $|V_x| = 400$  km/s.

In this study, BBFs are identified as fast Earthward flow channels exhibiting velocity  $V_x \ge 400$  km/s, according to the criteria previously established in the literature (Angelopoulos et al., 1992). Our analysis begins at simulation time t=400 s (top row), when no flows meet the selected criteria of  $V_x \ge 400$  km/s. However, we already observe patchy lines where  $B_z = 0$ , with an Earthward front extending longitudinally along the y-direction at a radial distance of approximately  $11~\rm R_E$ .

As time progresses, the contours of  $B_z=0$  spread in the azimuthal direction (compare Figures 1b, 1e, 1h and 1k) forming a topological X-line by t=500 s (the X - and O-line topologies are shown in Section 3.2 Figure 3). The formation of X-line, suggested by  $B_z=0$  proxy, is further supported by the diverging plasma flow. The first high-speed plasma flow with  $V_x\geq 400$  km/s emerges at t=450 s on the dusk side of the near-Earth magnetotail at an equatorial radial distance between 8 and 11  $R_E$  (see Figure 1d). This indicates an earlier onset of reconnection on the duskside -likely due to the large-scale influence of Hall physics, as previously shown in global hybrid PIC simulations (Lu et al., 2016). A tailward outflow with  $V_x\leq$  -400 km/s is clearly observed, with associated flow divergence region extending to a radial distance of approximately 11  $R_E$ . We focus only on the Earthward propagating BBF, as it has a direct impact on the ionosphere. This BBF persists for about 350 s, after which it interacts with other BBFs and evolves into a more complex plasma flow structure. This is consistent with observational studies, which typically report BBF durations ranging from a few minutes to 10 minutes (Baumjohann, 1990; Angelopoulos et al., 1992).

The emergence of the Earthward propagating BBF is accompanied by significant enhancements in both  $B_z$  and the magnitude of the pressure gradient ( $|\nabla P|$ ) at its Earthward front. Examining the pressure gradient, we first identify a narrow layer on the Earthward side of the strong gradient at a radial distance of about 8  $R_E$ , which also coincides with a strong spatial gradient in the  $B_z$  as depicted by the compactness of the black contours. At radial distances between approximately 7 and 9  $R_E$ ,  $B_z$  values

Figure 1. Snapshots of zoomed equatorial magnetotail in Vlasiator simulation: the x-component of the bulk flow velocity  $V_x$  (a, d, g, j), the z-component of the magnetic field  $B_z$  (b, e, h, k), and the magnitude of the pressure gradient  $|\nabla P|$  (c, f, i, l) at simulation times t = 400 s, t = 450 s, t = 500 s, and t = 550 s (from top to bottom). The black contour lines on all panels represent  $B_z$ , whereas the white contour lines indicate the locations where  $B_z$ =0. The solid and dashed magenta contour lines mark the boundaries of fast flows with  $V_x \ge 400 \, \mathrm{km/s}$  and  $V_x \le -400 \, \mathrm{km/s}$ , respectively.

range from 20 to 50 nT. In contrast, the background  $B_z$  in the magnetotail is generally below 10 nT (see panels (b, e, h and k) of Figure 1), but increases to approximately 40 nT at the front of the BBF. We interpret this region of enhanced pressure as the transition between the plasma sheet and the inner magnetosphere, where the dipole field dominates.

Figures 1g–i and Figures 1j–l show the state of the plasma sheet at simulation times t=500 s and t=550 s, respectively. During this time interval, the BBF structure grows in size, penetrates deeper towards Earth, drifts azimuthally duskward and slowly expands towards dawn (see Figures 1g and 1j). It is also notable that the layer of pressure gradient enhancement associated with BBF is clearly present at 8  $R_{\rm E}$  at t=500 s, and almost vanishes by t = 550 s. Additionally, the white  $B_z=0$  contours, which are the X-line proxy, shift tailward on the duskside. Overall, the evolution of the BBF reveals a dynamic interplay between the magnetic topology and the fast flow structures.

Figure 2 shows  $V_x$  in zoomed-in boxes centred around the BBF to evaluate its extent in the equatorial plane (Figure 2a-b) and along the Z-direction (Figure 2c-d). The azimuthal scale of the BBF flow channel is approximately 2  $R_E$  at t=450~s, expanding to approximately 4  $R_E$  by t=550~s as it moves Earthward and drifts duskward. In contrast to the size in the XY plane, the size of the BBF structure in the XZ plane changes from approximately 0.5  $R_E$  to 1.5  $R_E$ . This difference suggests that the fast-flow channels expand more in the azimuthal direction than in the off-equatorial plane direction.

# 3.2 FACs associated with BBFs in the Magnetotail






In this section, we examine the flow vorticity associated to BBF to explore the connection between the BBF and FACs that couple the magnetosphere and ionosphere.

Figure 3 illustrates the magnetic conjugacy between the BBF and the flow vorticity  $\Omega_z = (\nabla \times \boldsymbol{V})_z$  in the magnetotail current sheet and with the enhancement of the FAC density in the ionosphere at simulation time t = 550 s. The magnetic X- and O-lines (Alho et al., 2024) outline the magnetic topology, with the dusk X-line supporting magnetic reconnection at radial distances of 11–14  $R_E$  in the current sheet, as further evidenced by diverging plasma flow as indicated by the red contour line  $(V_x = 0)$  (see also Figure 2b). The black arrows represent the horizontal velocity components  $(V_{xy})$  in the current sheet. They clearly show strong Earthward and tailward plasma flows on the duskside of the current sheet than on the dawnside. The arrows also show the flow reversal near the X-line where the X-line fits well with  $V_x = 0$  (compare the X-line with the red contour line).

A direct consequence of the magnetic reconnection occurrence is the emergence of Earthward and tailward flows of BBFs near the reconnection points (see Supplementary Material Animation Movie 1: MovieS1.mov). As the BBF moves toward Earth, it induces a clockwise (counterclockwise) flow vortex viewed in the direction of the field lines from the tail side on the dawn (dusk) sides of it (see Figure 3). Note that on the dawnside flank of the BBF, the clockwise vortical flow develops into a closed vortex (see the velocity vectors at the most earthward flank), while the counterclockwise vortical flow on the duskside flank does not appear to form a fully closed structure. These vortical flows generate FACs flowing out of (into) the current sheet on the dawn (dusk) sides of the BBF. The blue (upward FAC) and red (downward FAC) traced magnetic field lines are centred at a radial distance of  $9\,R_{\rm E}$  and  $7.9\,R_{\rm E}$ , respectively, in the magnetotail current sheet. The generation of FACs due to vortical flow is in line with previous suggestions by e.g., Birn et al. (2004) that flow vortices and associated magnetic shear generate

Figure 2. (a, b) Same as Figures 1d and 1j at simulation times t = 450 s and t = 550 s, respectively, but showing only the duskside magnetotail in the XY plane at Z=0. (c, d) Snapshots of the x-component of the flow velocity  $V_x$  in the XZ plane at Y = 5.7  $R_E$  and Y = 6.3  $R_E$  as indicated by the red vertical lines on panels (a) and (b)

Figure 3. Magnetic conjugacy of the plasma flow vorticity  $\Omega_z = (\nabla \times \mathbf{V})_z$  in the magnetotail current sheet (where  $B_x = 0$ ) and field-aligned currents that are generated on the duskside and dawnside of the BBF (shown by blue and red coloured traced magnetic field lines), with the distributions of the R1/R2 FAC densities near the inner boundary, which corresponds to the FACs at ionospheric altitude at simulation time  $t = 550\,\mathrm{s}$ . The black arrows on the current sheet indicate the horizontal component  $(V_{xy})$  of the plasma velocity. The red contour line shows the plasma flow reversal  $V_x = 0$  between the Earthward and tailward flow regions. The reddish-orange and bluish-cyan block lines, respectively, on the current sheet indicate the magnetic X- and O-topologies from the Alho et al. (2024) FOTE method. The magenta line and the black arrow at the top right corner mark the boundary of the BBF structure where  $V_x = 400\,\mathrm{km/s}$  and the  $V_{xy}$  velocity vectors plotted on the current sheet, respectively.

field-aligned currents. As clearly seen in Figure 3, the clockwise (counterclockwise) flow vortices are mapped to upward (downward) FACs in the magnetotail. These FACs are connected to the R2 and R1 FACs at  $5.6\,\mathrm{R_E}$ , which are down-mapped to the ionospheric altitude. In addition to the stronger R1/R2 FAC pairs at the earthwardmost part of the BBF, weaker upward (duskside) and downward (dawnside) FACs are also evident along the flanks. These are associated with counterclockwise and clockwise vortical flows in the plasma sheet, respectively.

#### 3.3 Ionospheric signatures






In this section, we present the ionospheric signatures of a BBF as manifested by changes in the magnitudes of FACs, Hall conductance, proton and electron precipitation fluxes. To clearly illustrate the bijection between the occurrence of the BBF and its effects on the ionospheric variables, we also present the time evolution of the BBF structure, both at the magnetotail equatorial plane and its projection onto the ionosphere.

Figure 4 shows  $V_x$  in the magnetotail equatorial plane (a, g, m, and s), the projection of BBF structures onto the Northern Hemisphere ionosphere (b, h, n, and t), FACs (c, i, o, and u), Hall conductance (d, j, p, and v), the mean precipitating proton energy (e, k, q and w) and precipitating electron energy flux (f, l, r and x) in the Northern Hemisphere polar ionosphere at simulation times t = 400 (a -f), 450 (g -l), 550 (m-r), and 570 (s-x). The BBF projections are obtained by following the magnetic field lines from the magnetotail equatorial plane to the ionosphere.

At the simulation time t=400 s (top row), a high-speed Earthward plasma flow ( $V_x \ge 400$  km/s) has not yet emerged in the equatorial plane of the magnetosphere, while the ionospheric condition shows steady behaviour characterised by the classic Region 1 (R1) and Region 2 (R2) FAC patterns (Iijima and Potemra, 1978) where R1 FACs flow downward in the dawn sector, upward in the dusk sector and R2 FACs flow upward in the dawn sector and downward in the dusk sector. This is also observed in the uniform MLT distributions of the Hall conductance (Figure 4d) and precipitating proton and electron energies(Figure 4e and Figure 4f, respectively).

Figure 4h shows the projection of a BBF structure (see Figure 4g) from an equatorial radial distance between 8 and 11  $R_{\rm E}$  in the magnetotail onto the ionosphere. In the ionosphere, the projected BBF structure is stretched out in the east-west direction while covering a very limited latitudinal range in the north-south direction. However, at this time, no significant ionospheric signatures are seen in the FACs, Hall conductance, and precipitation energy values (Figures 4k and 4l). At  $t=450~{\rm s}$ , a BBF is indeed present; however, it is too weak to generate significant vorticity, which in turn limits the strength of FACs and their manifestation as an ionospheric signature.

During subsequent simulation times, the BBF increases in size and penetrates deeper toward Earth (bottom row). Visually, the most noticeable change in the FAC magnitude becomes apparent at the simulation time t = 485 s and beyond (see also Supplementary Material Animation Movies: MovieS1.mov and MovieS2.mov). At t = 485 s, the longitudinal size of the BBF structure increases (covering a wide range of MLTs), the R1/R2 FAC densities increase, and a bulged FAC structure forms between 21 and 23 MLT. Associated with the enhancement of the FACs, the magnitudes of the conductance, of precipitating proton energy and of precipitating electron integrated energy flux also increase. In subsequent simulation times, the FAC

Figure 4. Time evolution of the BBF in the magnetotail equatorial plane (a, g, m, s), projection of the BBF structure onto the ionosphere (b, h, n, t), FAC density at ionospheric altitude (c, i, o, u), ionospheric Hall conductance (d, j, p, v), precipitating proton mean energy  $E_{\rm pp}$  (e, k, q, w) and precipitating electron energy flux  $W_{\rm pe}$  (f, l, r, x) at simulation times t = 400 s (top row), t = 450 s (second row), t = 550 s (third row), and t = 570 s (bottom row). The ionospheric polar plots are shown on Magnetic Local Time (MLT) by Magnetic Latitude (MLAT) grid. The white contour lines in the magnetotail equatorial plane represent the  $B_z = 0$ , and the magenta contour lines indicate the locations of the BBFs with speeds  $V_x \ge 400\,\mathrm{km/s}$  (solid lines) and  $V_x \le -400\,\mathrm{km/s}$  (dashed lines). The blue and red contour lines on all polar plots denote upward and downward FACs, respectively, with amplitude  $\ge 0.1\mu\mathrm{Am}^{-2}$ .

magnitude continued to increase, and the bulge structure moved westward toward the duskside. The westward drift of the enhanced structures coincides with the duskward drift of the BBF in the near-Earth magnetotail.



Figure 5. Distribution of FACs (a–c), ionospheric Hall conductance  $\Sigma_{\rm H}$  (d–f), and precipitating electron energy flux  $W_{\rm pe}$  (g–i) over the Northern Hemisphere polar ionosphere. Panels (a, d, g) and (b, e, h) correspond to the Vlasiator simulation results at times t = 400 s and t = 550 s, respectively. Panels (c, f, and i) show the residual values ( $\delta {\rm FAC}$ ,  $\delta \Sigma_{\rm H}$  and  $\delta W_{\rm pe}$ ) obtained by subtracting the values at t = 400 s from those at t= 550 s. The contour lines in panels (a) and (b) illustrate the ionospheric electric potential at 4 kV intervals, and the numerical values at the bottom-right corner indicate the value of the cross-polar cap potential ( $\Delta \Phi$ ) for each simulation time. The grey O and X markers in all panels indicate the ionospheric projections of the seed points of the blue (duskside) and red(dawnside) traced field lines, respectively, shown in Figure 3. The magenta contour lines in the middle and right panels show the ionospheric projection of the boundary of the BBF at t= 550 s.

To quantify the effect of the BBF on the ionospheric observables more clearly, we compare the values during two simulation times: before (t=400 s) and after (t=550 s) the emergence of the BBF. Figure 5 presents a comparison of FAC density, ionospheric Hall conductance, and precipitating electron energy flux at two simulation times: t=400 s (left column) and t=550 s (middle column). The differences in the magnitudes of these variables are shown in the right side column. The magenta contour lines in the middle and right panels show the map of the BBF boundary at t=550 s (compare with Figures 4m and 4n). The grey O and X markers in all panels indicate the location of the traced field lines from magnetotail to ionosphere (refer to Figure 3). Figures 5a, 5d and 5g show the background values of FAC, Hall conductance and energy flux uniformly distributed over all MLT sectors and between  $64^{\circ}$ – $70^{\circ}$  MLATs. At this simulation time (t= 400 s), the cross-polar cap potential  $\Delta\Phi$  is 16

**Figure 6.** Time evolution of BBF signatures. Panels (a–d) show the original FACs at t = 500, 510, 520, and 530. Panels (e–g) present the residual FACs obtained by subtracting the FAC pattern at t = 500 from the FACs in the second row. Panels (h–j) and (k–m), respectively, show the residual Hall conductances and precipitation energy fluxes obtained by subtracting the values at t = 500 s from the values at t = 510, 520, and 530 in the same order as FACs.

kV. At simulation time t=550 s (middle column), the  $\Delta\Phi$  value increases to 19 kV, and the magnitudes of R1/R2 FACs, Hall conductance, and energy flux show significant enhancement specifically around 20-21 MLT. Panels (c, f, and i) show the changes during the two simulation times obtained by subtracting the values at t=400 s from the corresponding values at t=550 s. The changes in the values are stronger around the 21 MLT sector and clearly indicate that the enhancements in FAC, Hall conductance, and precipitation energy flux at simulation time t = 550 s are all associated with the BBF.


We further compare the MLT/MLAT locations of the enhancements with the location of the projections of the BBF boundary (magenta contour line) and the clockwise (counterclockwise) flow vortices indicated by the grey X (O) markers for simulation time t=550 s. The BBF boundary maps around  $69^{\circ}$  MLAT and between 20:30-22:45 MLTs, while the clockwise (counterclockwise) vortices (indicated by X and O markers, respectively) map at about 21:10 (22:45) MLTs and within  $67^{\circ}$ - $69^{\circ}$  MLATs. The

observed enhancements in FAC, conductance and energy flux map mainly between 20-21 MLTs (see middle column). This is also seen clearly on the residual plots (right column).

The peaks of the R2/R1 FAC enhancements (see Figures 5b and 5c) coincide with the MLT-MLAT locations of the X (O) markers, further supporting the notion that the source regions of the downward R2 (upward R1) FAC enhancement are the vortical plasma flows at the Earthward flanks of the BBF in the magnetotail. On the other hand, the peaks of the Hall conductance and precipitation energy flux enhancements are located between the R2/R1 peaks, occurring around 20:30 MLT and equatorward of the BBF boundary between 65° and 67° MLATs (see panels: e, f, h and i).

Figure 6 illustrates the time evolution of BBF signatures, as manifested by changes in the magnitudes of the FACs, Hall conductance, and precipitation energy flux. The residual values are calculated by subtracting the values at t = 500 from those at t = 510, 520, and 530. The contour lines over the residual conductance and energy flux plots represent 10% of the residual downward (red lines) and upward (blue lines) FACs shown in panels (e–f), respectively. All FACs in the first and second rows display the typical FAC distribution patterns, with high-latitude R1 FACs flowing downward in the dawn sector and upward in the dusk sector, and lower-latitude R2 FACs exhibiting the opposite polarity. At all times, the magnitude of the R1/R2 FACs is stronger on the duskside than on the dawn side of the ionosphere, corresponding to the emergence of the BBF on the duskside of the magnetotail.

The time evolution of the ionospheric signatures is evident in the residual patterns. In the residual FAC patterns (panels e–g), an enhanced pair of FACs emerges between 20 and 22 MLT at magnetic latitudes between 65° and 70°, with downward currents in the east (midnight side) and upward currents in the west (dayside). The residual FAC observed at t=510s (Figure 6e) continues to increase in magnitude over time, expanding toward the dayside (see Figure 6g). In line with the appearance of residual FACs, there is also an increase in the residual values of conductance (fourth row) and precipitation energy fluxes (bottom row). The residual values of conductance and energy flux peak at MLATs between 65° and 70°, and between 20 and 21 MLTs.

# 3.4 FAC closure in the Ionosphere





Figure 7 presents the ionospheric electric field, electron drift velocity, and Pedersen and Hall currents, and the closure of a pair of magnetospheric FACs through ionospheric currents at simulation time t =550 s. As discussed in Section 3.3, enhanced R1 and R2 FACs associated with BBFs in the magnetotail occurred in the evening MLT sector. Here, these FACs are shown in Figure 7e. Unlike the well-known configuration of FACs in the high-latitude ionosphere, where R1 (R2) FACs are located in the poleward (equatorward) part of the auroral oval, this enhanced pair of FACs is aligned in the dawn-dusk (east-west) direction. More specifically, it is along the northwest-southeast direction rather than in the north-south direction. Below, we discuss the closure of these enhanced FACs in the ionosphere.

Figure 7a shows the distribution of the ionospheric electric field vectors in the evening MLT sector. As discussed in Section 2.2, this electric field is calculated from the gradient of the ionospheric electric potential. In the MLT sector, between 18 and 20, the electric field is entirely poleward. Under normal conditions, this part of the electric field drives the Pedersen current, which connects the downward R2 FAC with the upward R1 FAC on the dusk side of the ionosphere and also drives the eastward

component of the ionospheric horizontal current, commonly referred to as the eastward electrojet (Workayehu et al., 2019). However, in the 20–22 MLT sector, where there is an enhancement of the FACs associated with the BBF, the magnitude of the electric field increases, and its direction rotates toward the west. This is consistent with the east-west alignment of the enhanced FACs.

Figure 7b shows the distribution of the electron drift velocity vectors. Between 20 and 21 MLT, where the electric field rotated to the east-west direction, an equatorward flow channel with an associated duskward and dawnward flow rotation occurs as shown by black arrows. The flow channel in the ionosphere is an image of the Earthward plasma flow of the BBF, whereas the duskward and dawnward rotational flows are related to the counterclockwise and clockwise vortical flows observed in the magnetotail (see Figure 7e). It is important to note that an enhanced ionospheric flow channel is not observed along the entire magnetospheric BBF channel. Strong ionospheric signatures arise mainly at the flanks of the BBF, where flow shear generates FACs that couple efficiently into the ionosphere. In contrast, the central part of the BBF couples only weakly, and the visibility of the flows is further controlled by local ionospheric conductivity. As a result, the relatively broad BBF in the plasma sheet is typically associated with localised ionospheric flow channels rather than wide, uniform enhancements. Overall, the plasma flow channels in the magnetotail and in the ionosphere are consistent with each other, and our results are in line with a schematic illustration of the connection between the magnetospheric convection pattern and the ionospheric equivalent current pattern by e.g., Kauristie et al. (2000).

Figures 7c and 7d show the distributions of the Pedersen ( $J_{Pedersen}$ ) and Hall ( $J_{Hall}$ ) currents, respectively.  $J_{Pedersen}$  flows in the direction of the electric field, whereas  $J_{Hall}$  flows perpendicular to both electric and magnetic fields. Around 20:30 MLT, both  $J_{Pedersen}$  and  $J_{Hall}$  are enhanced and in line with the direction of the enhanced electric field (see Figure 7a)  $J_{Pedersen}$  flows in the north-west direction as indicated by the red dotted arrow, while the enhanced  $J_{Hall}$  flows north-east, opposite to the equatorward electron flow direction. This is an MLT sector where an enhanced downward R2 and upward R1 FACs flow and indicates that the two enhanced FACs are mainly connected by  $J_{Pedersen}$ . Figure 7d clearly indicates that electron convection carries  $J_{Hall}$  current which is antiparallel to the plasma flow direction. The fact that the enhanced downward and upward FACs, generated by the emergence of bursty bulk flow, are connected by  $J_{Pedersen}$  resembles the mechanism of large-scale SCW closure in the ionosphere. However, in the case of the SCW, the cross-tail current is diverted into the ionosphere, forming an SCW, which consists of downward (upward) FACs on the dawn side (duskside) of the wedge and a westward electrojet in the ionosphere (Yao et al., 2012, and references therein).

Figure 7e shows the magnetic conjugacy of BBFs on the magnetotail current sheet and ionospheric FACs upmapped to  $5R_{\rm E}$  for better visualisation. As clearly seen, the upward FAC and clockwise plasma flow in the dawnside flank of the flow channel directly coincide with the downward R2 FAC and the eastward plasma flow in the ionosphere, while the downward FAC and the counterclockwise vortical flow in the magnetotail is conjugate to the R1 FAC and the westward vortical flow in the ionosphere. The ionospheric equatorward flow channel is directly linked to the Earthward flow of the BBF in the magnetotail.

Figure 7. (a) Ionospheric electric field E, (b) plasma drift velocity  $V_d$ , (c) Pedersen current  $J_{\rm Pedersen}$ , (d) Hall current  $J_{\rm Hall}$  and (e) magnetic conjugacy of ionosphere and FACs associated with BBF, and plasma flow pattern on the dawn- and dusksides of the BBF in the current sheet at simulation time t=550~s. The colournap on the current sheet shows the x-component of velocity.

# 305 4 Discussion







We investigate the evolution of a BBF in the magnetotail, focusing on its generation and movement toward Earth using a global 6D hybrid-Vlasov simulation. Our results indicate that magnetic reconnection is responsible for the formation of BBFs (Angelopoulos et al., 1992; Baumjohann, 1990). Overall, the results capture the general characteristics of BBFs and illustrate a dynamic interplay between magnetic topology and flow structures (see Supplementary Animation Move S1).

We examined the relationship between the BBF and FACs in the magnetotail through the analysis of flow vorticity in the near-Earth magnetotail. We found that as the BBF moves Earthward, it induces flow vortices around its Earthward flanks (see Figure 3 and Supplementary Movie S1). We further demonstrated that the locations of these flow vortices coincide with a pair of FACs flowing out of (into) the plasma sheet on the dawnside (duskside) flanks of the BBF. These findings are in agreement with previous findings that flow vortices and associated magnetic shear play a significant role in the generation of FACs (Birn et al., 2004; Yu et al., 2017).

The ionospheric signatures of BBF are observed as enhancements in various ionospheric variables, including FACs, ionospheric currents, conductances, precipitating proton energy, and precipitating electron integrated energy flux. These signatures also include the formation of localised ionospheric plasma flow channels and the westward drift of these features, which correlates with the duskward movement of the BBF within the magnetotail.

Specifically, we found that the magnitudes of the R1/R2 FACs significantly increase in the pre-midnight MLT sector, as discussed in Section 3.3. This increase is attributed to the FACs generated by BBFs in the magnetotail. In these same MLTs and latitude ranges, we also observed an increase in the precipitating proton average energy and precipitating electron integrated energy flux, as well as in conductances, and a southwest-directed plasma flow channel. In Section 3.4, we demonstrated that the enhanced FACs are closed in the ionosphere by Pedersen currents, which flow along the electric field. These findings align with previous studies, including those by Pitkänen et al. (2011); Juusola et al. (2013); Yu et al. (2017); Sergeev et al. (2020); Wei et al. (2021); Ferdousi et al. (2021); Grandin et al. (2023).

Recent studies by Pitkänen et al. (2011) and Juusola et al. (2013) examined the ionospheric signatures of BBFs during quiet and active geomagnetic conditions, respectively. Utilising data from the Cluster spacecraft and the EISCAT radar, along with several ground-based measurements, Pitkänen et al. (2011) found that BBFs are linked to auroral streamers and localised equatorward plasma flows in the ionosphere. Juusola et al. (2013) similarly investigated the formation of Earthward fast flows in the plasma sheet and the associated ionospheric signatures during substorm onset. Their findings revealed that the Earthward fast flow is connected to equatorward-propagating auroral streamers, an enhanced poleward equivalent current, and a westward travelling surge. Our simulation results, which show an intensified southwestward (see Figure 7b) flow channel accompanied by a northeastward Hall current (see Figure 7d), as well as the westward drift of the enhanced R1/R2 FACs, precipitating proton and electron energies, and ionospheric conductivities, align with the findings of Pitkänen et al. (2011) and Juusola et al. (2013).

Using global MHD simulations, Ferdousi et al. (2021) reported on auroral structures associated with multiple BBFs in the magnetotail. Their results indicate that each of the auroral structures is associated with a pair of FACs of opposite polarity. By mapping these BBF structures to the ionosphere, the authors demonstrated that BBFs originating from the flanks of the

magnetotail align in an east-west direction and are narrow in latitude but wider in longitude. These findings are consistent with our observation that the ionospheric signature of a BBF appears as an enhancement in the R1/R2 FACs, and that the projection of the BBF structure aligns in the east-west direction as well.

Using the inertialized Rice Convection Model (RCM-I) simulation, Wei et al. (2021) investigated the magnetospheric driver of the westward drifting auroral bulge structure (Westward Travelling Surge, WTS) in the high-latitude ionosphere. The authors demonstrated that a plasma-sheet bubble induces a localised upward FAC on the duskside of the bubble, leading to the formation of the WTS in the ionosphere. They concluded that the duskward expansion of the bubble in the magnetosphere is responsible for the westward surge of the auroral bulge structure in the ionosphere. In our simulation, we observed upward FACs on the duskside of the BBF, along with the westward drift of the ionospheric signatures of BBFs, which aligns with the findings of Wei et al. (2021).

Using MHD simulation coupled with a kinetic ring current model, Yu et al. (2017) investigated the effect of BBFs on global-scale current systems. The authors demonstrated that vorticity induced by BBFs at the inner magnetosphere generates a localised pair of FACs that are observed as ionospheric signatures of BBFs. This is in line with our observations. However, unlike our results, which show an upward R1 (and a downward R2) pair of FACs associated with a BBF, their study indicates R1-sense upward and downward FAC pairs that close within the ionosphere. This difference can be attributed to an offset in the radial distance of the vorticities in the dawn- and dusk-sides of the BBF from the centre of the Earth. Notably, we have observed the dawnside vorticity (mapped to R2 FAC) at about  $7.9 R_{\rm E}$  and the duskside vorticity (mapped to R1 upward FAC) at about  $9 R_{\rm E}$  from the center of the Earth (see the origins of the traced field lines in Figure 3).

Recently, Grandin et al. (2023) examined proton precipitation fluxes using 3D-3V hybrid-Vlasov simulations. Their simulation setup and the solar wind driving conditions are the same as those used in this study, with the only difference being the inner boundary condition. In their simulation, the inner boundary was a near-ideal conducting sphere, whereas in the current simulation, the inner boundary is an ionospheric model. The authors identified a region of increased proton precipitation fluxes in the nightside ionosphere, which they attributed to the dynamics of BBFs in the magnetotail. This finding is consistent with our observations of enhanced energies in precipitating protons and electrons.

# 5 Summary and Conclusion








We present the ionospheric signatures of BBFs utilising a global 6D hybrid-Vlasov simulation coupled with an ionospheric model. The FAC density is computed near the inner boundary from the curl of the simulation magnetic field, and the BBFs are identified as fast Earthward flow channels with  $V_x \ge 400\,\mathrm{km/s}$ . Both the FACs and the BBF structures are traced down to the ionospheric altitude along the magnetic field lines. We analyse changes in the magnitudes of FAC densities, ionospheric conductances, precipitating proton energy and precipitating electron integral energy flux, as well as alterations in the directions of horizontal currents, electric fields, and ionospheric flow velocity vectors, and use them as the ionospheric manifestations of BBFs.

The most important findings of this study are as follows:

- 1. Following magnetic reconnection at a radial distance between about 10– $12~R_{\rm E}$  in the magnetotail current-sheet, a BBF with  $V_{\rm x} \geq 400$  km/s is ejected Earthward on the dusk side of the magnetotail. As this fast flow approaches Earth, its flow turns azimuthally duskward, while slowly expanding dawnward. This interaction induces flow vortices on the duskside and dawnside of the flow (see Supplementary Movie S1).
  - 2. The dawnside (duskside) vortices generate a pair of FACs flowing out of (into) the magnetotail current sheet; these currents, when mapped into ionospheric altitude, coincide with the enhanced R2 (R1) FACs that flow into (out of) the pre-midnight ionosphere, respectively.
- 3. The ionospheric signature of BBFs is manifested by enhanced structures in FACs, ionospheric conductances, precipitation energy flux, and both Pedersen and Hall currents, as well as the presence of localised ionospheric plasma flow channels.
- 4. The enhanced R2/R1 FACs at ionospheric altitude are closed by the north-westward flowing Pedersen current.
- 5. The duskward motion of the BBFs in the magnetotail is linked to the westward drifts of the ionospheric signatures.
- 6. The mapping of BBF structure to the ionosphere shows that it is predominantly aligned in the east-west direction (see Figure 4(h, n, t)), while the flow channel is north-south directed (see Figure 7b).
- In general, our 6D hybrid-Vlasov simulation results are consistent with most of the results from previous studies, confirming that the ionospheric solver works reasonably well and that the findings of the study are reliable.
  - . The Vlasiator simulation code (Pfau-Kempf et al., 2024) is distributed under the GPL-2 open source license. The simulation data used in this study is stored at (Suni and Horaites, 2024). The Analysator Python package (Battarbee et al., 2021) and the open-source VisIt software (Childs et al., 2012) were used for data analysis and visualisation.
- 390 . Two Supplementary Animation Movies "MovieS1.mov" and "MovieS2.mov" are presented to supplement Figures 3 and 4.
  - . AW and MP conceptualised the study. AW analysed the simulation data, performed the visualisations, and wrote the original draft. MA assisted with the visualisation. AW, MP, LJ, MG, IZ, MA, VK and HK contributed to the interpretation of the results. UG, YP, MB and JS performed the simulation runs utilised in this study. MP is the Vlasiator PI and supervisor of this study. All co-authors reviewed, commented and edited the paper.
- 395 . An author is a member of the editorial board of Annales Geophysicae.


. AW and MP acknowledge the Research Council of Finland (grant numbers: 347795-HISSA and 352846-FORESAIL).


The work of MB and UG are supported by the EuroHPC "Plasma-PEPSC" Centre of Excellence (grant number 4100455) and the Research Council of Finland matching funding (grant number 359806). MB acknowledges the Research Council of Finland grant number 352846.

MG acknowledges funding from the Research Council of Finland (grant 360433-ANAON). YP acknowledges funding from the Research 400 Council of Finland (grant 339756-KIMCHI).

MA acknowledges the Research Council of Finland grant numbers 352846 and 361901, and the Inno4Scale project via European High-Performance Computing Joint Undertaking (JU) under Grant Agreement No 101118139. The JU receives support from the European Union's Horizon Europe Programme. The work of MA is also funded by the European Union (ERC grant WAVESTORMS - 101124500). Views and opinions expressed are, however, those of the author(s) only and do not necessarily reflect those of the European Union or the European Research Council Executive Agency. Neither the European Union nor the granting authority can be held responsible for them.

The work of JS was made possible by a doctoral researcher position at the Doctoral Programme in Particle Physics and Universe Sciences funded by the University of Helsinki.

The authors thank the Finnish Computing Competence Infrastructure (FCCI), the Finnish Grid and Cloud Infrastructure (FGCI) and the University of Helsinki IT4SCI team for supporting this project with computational and data storage resources. The authors wish to acknowledge CSC – IT Center for Science, Finland, for computational resources. The simulation presented in this work was run on the LUMI-C supercomputer through the EuroHPC project Magnetosphere-Ionosphere Coupling in Kinetic 6D (MICK, project number EHPC-REG-2022R02-238).

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
