# Peer review of "Ionospheric signatures of a Bursty Bulk Flow in the 6D Vlasiator simulation"

_EGUsphere, 2025_

## Author Comment (AC1)

**Title: Ionospheric signatures of Bursty Bulk Flows in the 6D Vlasiator simulations**

Authors: Abiyot Bires Workayehu, +co-authors

**Replies to Referee #2**

We thank Referee #2 for comments and constructive suggestions on our manuscript. Below, we provide point-by-point responses to all comments. The referee's comments are presented in black, while our replies are shown in blue directly below each comment. We have carefully considered all suggestions and, where appropriate, will make corresponding revisions in the manuscript. We hope our answers and the corresponding suggested changes (where necessary) in the manuscript are satisfactory.

This paper presents numerical simulation results from the 6D hybrid-Vlasov code Vlasiator, coupled with an electrostatic ionospheric model, to investigate the ionospheric signatures of bursty bulk flows (BBFs). The study's core contribution lies in the detailed mapping of BBF-induced vortical flows to specific field-aligned current (FAC) systems and their associated ionospheric responses. These include enhancements in FACs, ionospheric conductances, and precipitating particle energies. The manuscript is well-written, and the results are clearly presented. The manuscript is clearly worth publishing, but before I could recommend publication, the authors could address a few minor comments, which are listed below.

1. Comment (1): The simulation uses steady and extreme solar wind conditions (Vsw = 750 km/s, Bz = -5 nT). How representative are these results of more typical solar wind conditions? Please justify this choice and, importantly, discuss how the results might change under more typical or variable solar wind conditions. For example, would a weaker IMF or lower solar wind velocity still produce such distinct ionospheric signatures?

   We thank the reviewer for this comment. Yes, the solar wind driving conditions used in the Vlasiator simulation correspond to a fast solar wind stream (Vsw = 750 km/s) with moderately active conditions (Bz = –5 nT). The fast solar wind was chosen to speed up the initialisation phase of the simulation run. Importantly, this kind of simulation would be significantly more expensive computationally, and the present Vlasiator runs with these solar wind driving conditions are the best that can be performed with the current supercomputers.

   To clarify our choice, we propose adding the following text in Section 2.1: "The fast solar wind was chosen to speed up the initialisation phase of the simulation run (Palmroth et al., 2023), while Bz = –5 nT represents conditions favourable for magnetic reconnection without being strongly disturbed."

   A stronger southward IMF and higher solar wind speed increase the coupling between the solar wind and Earth's magnetosphere, leading to more frequent substorms and BBFs in

the magnetotail (Zhang et al., 2016), and the same processes are expected during weaker conditions but with lower amplitude and lower occurrence rates (Zhang et al., 2016). Regarding ionospheric signatures, generally, stronger solar wind driving conditions result in more significant ionospheric effects and more pronounced signatures. A weaker solar wind driving may still produce less intense signatures, but not necessarily less distinct ones.

2. Comment (2): The BBF criterion ($Vx \geq 400$ km/s) is standard, but duration criteria are not discussed. How long does this BBF persist?

We thank the reviewer for this comment. The BBF persists for about 350 s, after which it interacts with other BBFs and evolves into a more complex plasma flow structure.

We propose adding the following text in Section 3.1 of the manuscript: "The BBF persists for about 350 s, after which it interacts with other BBFs and evolves into a more complex plasma flow structure. This is consistent with observational studies, which typically report BBF durations ranging from a few minutes to 10 minutes (Baumjohann et al., 1990; Angelopoulos et al., 1992)."

3. Comment (3): The study focuses on a single BBF event within the simulation. Is it a typical or ideal BBF produced by the Vlasiator model? Have the authors observed similar signatures for other BBFs in their simulations?

The BBF analysed here is a typical BBF produced by the Vlasiator simulation. Yes, BBFs with more complex structures and comparable ionospheric signatures are also observed in the later stages of the simulation.

4. Comment (4): The duskside preference of BBFs is noted. How does this asymmetry influence the ionospheric signatures compared to dawnside BBFs?

This paper discusses signatures of a duskside BBF, and we have not observed an isolated dawnside BBF in the simulation to make a one-to-one comparison of ionospheric signatures. Therefore, it is difficult to draw conclusions about the signature of a dawnside BBF compared to the duskside. However, as shown in the present paper, the dusk preference of BBFs causes ionospheric signatures to cluster in the pre-midnight MLT sector, where we have observed enhanced FAC, conductance, and flow channel, while the post-midnight ionosphere displays weaker and spatially and temporally smooth distributions of these variables.

5. Comment (5): The simulation assumes a 0° dipole tilt, which is a simplification. Could including a more realistic dipole tilt affect ionospheric coupling?

Yes, including realistic dipole tilt angles can affect ionospheric coupling. Assuming a 0° tilt simplifies the geometry but neglects hemispheric and seasonal asymmetries that influence how the magnetosphere couples to the ionosphere. While using a 0° tilt angle roughly represents average equinoctial conditions, incorporating a realistic tilt changes the geometry of magnetopause reconnection, alters the mapping of field-aligned currents and convection into the ionosphere, and modifies ionospheric conductance through seasonal and diurnal variations in solar illumination.

6. Comment (6): The thin-shell approximation at 100 km altitude is justified, but how might altitude-dependent effects influence the results?

   We thank the reviewer for this comment. The thin-shell approximation effectively captures large-scale ionospheric current systems and electric fields through height-integrated Pedersen and Hall conductances. However, we do not know the influence that altitude-dependent ionospheric profiles may have on the results. High-resolution volumetric measurements of ionospheric parameters, such as those anticipated from the upcoming EISCAT3D (McCrea et al., 2015), will help to resolve this.

7. Comment (7): A recent study by Kumar et al. (2025) (https://doi.org/10.1029/2024JA032953), using THEMIS and MMS observations, reported a dawn-dusk asymmetry in flows and a significant deceleration of these flows earthward of X <-15 RE. Please comment on whether the Vlasiator simulation reproduces a comparable braking effect in the near-Earth region.

   The duskside preference and near-Earth deceleration of fast plasma flows observed in our study agree with Kumar et al. (2025), who observed significant flow braking earthward of X < -15 RE and more frequent occurrence in the premidnight region than postmidnight. However, the detailed braking effect reported by Kumar et al. is not explicitly addressed in this paper. Further study, directly comparing Vlasiator results with observations in this region, would be needed to assess whether the braking effect reported by Kumar et al. (2025) is reproduced.

8. Comment (8): On page 2, line 45, "breaking region" should likely be "braking region". Please check for this consistency throughout the manuscript. Yes, the reviewer is correct. The term will be corrected to "braking region", and the manuscript will be carefully checked for consistency throughout.

9. Comment (9): Line 199-200: At t = 450 s, a BBF is active, yet the paper says there are "no significant ionospheric signatures." What is the expected delay between BBF arrival and ionospheric response in this setup?

   Yes, at $t = 450$ s, a BBF is indeed present; however, it is too weak to generate significant vorticity, which in turn limits the strength of FACs and their manifestation as an ionospheric signature. The expected delay between the BBF and the ionospheric response at this simulation time is approximately 18s. This estimate is based on the Alfvén speed along the field line connecting the BBF location to the coupling radius at 5.6 $R_E$, with an additional 2 s delay from the coupling radius to the ionosphere (Ganse et al., 2025) to obtain the total delay time.

   We propose adding the following clarification to the manuscript: "At $t = 450$ s, a BBF is indeed present; however, it is too weak to generate significant vorticity, which in turn limits the strength of FACs and their manifestation as an ionospheric signature."

10. Comment (10): The use of Bz = 0 as a proxy for reconnection lines is an oversimplification, as it does not inherently confirm the presence of active reconnection. Could the authors elaborate on how additional reconnection indicators—such as flow reversals, Hall magnetic field signatures, or localised energy conversion— align with the Bz = 0 regions in their simulation?

    We thank the reviewer for highlighting the limitations of using $B_z = 0$ as a proxy for reconnection lines. We fully agree that $B_z = 0$ alone does not confirm active reconnection.

In our study, $B_z = 0$ was used in Section 3.1 as a simplified indicator of potential X-line locations. To identify actual reconnection sites, we additionally applied the method by Alho et al. (2024), which distinguishes both X- and O-lines. This approach has been used in previous publications and provides a more robust and reliable identification of reconnection points in Vlasiator simulations. As shown in Figure 3 and Supplementary Movie S1, the X-lines obtained using this method reasonably align with the flow reversal at points where active reconnections occur.

**References**

Alho, M., Cozzani, G., Zaitsev, I., Kebede, F. T., Ganse, U., Battarbee, M., Bussov, M., Dubart, M., Hoilijoki, S., Kotipalo, L., Papadakis, K., Pfau-Kempf, Y., Suni, J., Tarvus, V., Workayehu, A., Zhou, H., and Palmroth, M.: Finding reconnection lines and flux rope axes via local coordinates in global ion-kinetic magnetospheric simulations, Annales Geophysicae, 42, 145–161, https://doi.org/10.5194/angeo-42-145-2024, publisher: Copernicus GmbH, 2024.

Angelopoulos, V., Baumjohann, W., Kennel, C. F., Coroniti, F. V., Kivelson, M. G., Pellat, R., Walker, R. J., Lühr, H., and Paschmann, G.: Bursty bulk flows in the inner central plasma sheet, Journal of Geophysical Research: Space Physics, 97, 4027–4039, https://doi.org/10.1029/91JA02701, _eprint: https://onlinelibrary.wiley.com/doi/pdf/10.1029/91JA02701, 1992.

Baumjohann, W., Paschmann, G., and Lühr, H.: Characteristics of high-speed ion flows in the plasma sheet, Journal of Geophysical Research, 95, 3801–3809, https://doi.org/10.1029/JA095iA04p03801, 1990.

Ganse, U., Pfau-Kempf, Y., Zhou, H., Juusola, L., Workayehu, A., Kebede, F., Papadakis, K., Grandin, M., Alho, M., Battarbee, M., Dubart, M., Kotipalo, L., Lalagüe, A., Suni, J., Horaites, K., and Palmroth, M.: The Vlasiator 5.2 ionosphere – coupling a magnetospheric hybrid-Vlasov simulation with a height-integrated ionosphere model, Geoscientific Model Development, 18, 511–527, https://doi.org/10.5194/gmd-18-511-2025, publisher: Copernicus GmbH, 2025.

McCrea, I., Aikio, A., Alfonsi, L., et al.: The science case for the EISCAT_3D radar, Progress in Earth and Planetary Science, 2, 21, https://doi.org/10.1186/s40645-015-0051-8, 2015.

Palmroth, M., Pulkkinen, T. I., Ganse, U., et al.: Magnetotail plasma eruptions driven by magnetic reconnection and kinetic instabilities, Nature Geoscience, 16, 570–576, https://doi.org/10.1038/s41561-023-01206-2, 2023.

Zhang, L. Q., Baumjohann, W., Wang, C., Dai, L., and Tang, B. B.: Bursty bulk flows at different magnetospheric activity levels: Dependence on IMF conditions, Journal of Geophysical Research: Space Physics, 121, 8773–8789, https://doi.org/https://doi.org/10.1002/2016JA022397, 2016.

---

## Author Response (AR1)

**Title: Ionospheric signatures of a Bursty Bulk Flow in the 6D Vlasiator simulation**

Authors: Abiyot Bires Workayehu, +co-authors

**Replies to Referee #1**

We thank Referee #1 for comments and constructive suggestions on our manuscript. Below, we provide point-by-point responses to all comments. The referee's comments are presented in black, while our replies are shown in blue directly below each comment. We have carefully considered all suggestions and, where appropriate, made corresponding revisions in the manuscript. All changes in the manuscript are marked(added texts are shown in blue, and deleted texts are shown in red and struck through by a horizontal line). We hope our answers and the corresponding changes (where necessary) in the manuscript are satisfactory.

**General comments:**

In this paper, numerical simulation results from the 6D hybrid-Vlasov code Vlasiator are utilised to study the ionospheric signatures of a bursty bulk flow (BBF). This is the first time when an ionospheric model enabling a two-way magnetosphere-ionosphere coupling is used with Vlasiator. The magnetospheric signatures of the BBF include a flow channel of earthward fast plasma flow, which has a significant azimuthal orientation at the farther parts, and an appearance of oppositely directed vorticity on the flanks of the BBF channel. The flow vortices/vorticity induces field-aligned currents flowing into the ionosphere on the dawnside and flowing out from the ionosphere on the duskside of the BBF channel. The ionospheric signatures include a localised enhanced equatorward plasma flow corresponding to the earthward part of the magnetospheric BBF and signatures of enhanced FACs and vortical plasma flow on the flanks of the enhanced flow channel consistent with the magnetospheric counterpart. In addition, observable signatures in ionospheric conductances, precipitation energy flux and both Pedersen and Hall currents can be seen in association with the BBF.

This is the first time when ionospheric signatures of BBFs obtained from the Vlasiator are presented. Although the BBF in the simulation is generated rather close to Earth, the simulation produce many signatures reported in previous observational and simulation studies. This suggests that the simulation codes used can be utilised to study dynamical magnetosphere-ionosphere coupling and the results are reasonably well and can be used in comparison with observations. The manuscript is clearly worth publishing, but before I could recommend publication, the authors could address a few comments, which are presented below.

**Specific comments:**

1. Comment (1): The authors present only one BBF case. The authors could consider modifying the manuscript title to "Ionospheric signatures of a bursty bulk flow in the 6D Vlasiator simulation".

We thank the reviewer for the suggestion. We agree, and the manuscript title is modified to "Ionospheric signatures of a Bursty Bulk Flow in the 6D Vlasiator simulation."

2. Comment (2): When discussing the FAC pair, the authors focus now on the earthward-most part of the BBF. However, the signatures of weaker upward FAC on the duskside flank and downward FAC on the dawnside flank of the BBF are visible both in the magnetosphere and in the ionosphere also in the tailward (more dusk-dawn oriented) part the BBF (see e.g, the vorticities and FACs in Figure 3). The authors could consider adding some discussion about that.

The following text is added to Section 3.2 of the manuscript: "In addition to the stronger R1/R2 FAC pairs at the earthwardmost part of the BBF, weaker upward (duskside) and downward (dawnside) FACs are also evident along the flanks. These are associated with counterclockwise and clockwise vortical flows in the plasma sheet, respectively."

- 3. Comment (3): It seems that the vortical flows on the flanks of the BBF structure do not form complete vortices. The authors might want to point that out in the text.

  The following text is added to Section 3.2 of the manuscript: "Note that on the dawnside flank of the BBF, the clockwise vortical flow develops into a closed vortex (see the velocity vectors at the most earthward flank in Figure 3). In contrast, the counterclockwise vortical flow on the duskside flank does not appear to form a fully closed structure."
- 4. Comment (4): Line 19 and elsewhere: The use of brackets for the reference is a bit weird: Angelopoulos et al. (1992). Should it be written (Angelopoulos et al., 1992) here? Compare the citation style on line 24.
  We thank the referee for pointing this out. The reference has been updated to "(Angelopoulos et al., 1992)", and the manuscript has been reviewed to ensure consistent citation formatting throughout.
- 5. Comment (5): Lines 37-40: Discussion of Sergeev et al. (2020) on these line is not accurate. Sergeev et al. (2020) do not present any BBF observations in their paper. Also the last statement about the arcs tending to align with the direction of the electric field appears not to be based on Sergeev et al. (2020). Please, check again Sergeev et al. (2020) and rewrite this paragraph if you want to introduce Sergeev et al. (2020) work here.

We agree with the reviewer that Sergeev et al. (2020) do not present BBF observations. Instead, they discuss the potential link between auroral arcs and BBFs based on auroral arc observations. Now, the paragraph has been modified as follows: "Using a combination of observational data and empirical magnetospheric models, Sergeev et al. (2020) investigated the origins and orientations of nightside auroral arcs. They found that the majority of nightside arcs originate from the magnetotail current sheet region and argued that magnetospheric flow channels, such as BBFs, are the most likely source of these arcs. Furthermore, they demonstrated that structures, which appear nearly sun-aligned in the plasma sheet, become increasingly azimuthally aligned when mapped to the ionosphere."

6. Comment (6): Figure 3 (and Figure 7e): Is the length of the Vxy vector shown on the top of the Figure 3 on the right hand side of the magenta text "Vx = 400 km/s" 400 km/s?

The magenta line at the top right corner of Figures 3 and 7e marks the boundary of the BBF structure where  $V_x = 400 \text{ km/s}$ . The black arrow adjacent to it represents the  $V_{xy}$  velocity vectors plotted on the current sheet. Now, we clarify this distinction by adding the following text in Figure 3 caption: "The magenta line and the black arrow at the top right corner mark the boundary of the BBF structure where  $V_x = 400 \text{ km/s}$  and the  $V_{xy}$  velocity vectors plotted on the current sheet, respectively."

7. Comment (7): Line 223: Should the clockwise vortex be indicated by X and counter-clockwise by O? Compare e.g. to lines 224-225. Double-check and correct the text if necessary.

Thank you for pointing that out. Yes, O and X markers correspond to counterclockwise and clockwise vortices, respectively. The text manuscript has been updated accordingly.

8. Comment (8): Line 305: Do you mean here southwest-directed plasma flow channel in the ionosphere? You write southeast-directed plasma flow channel.

Yes, thank you for pointing this out. The text has been updated to "southwest-directed plasma flow channel in the ionosphere".

9. Comment (9): Lines 355-356: On lines 149-151 the authors describe the general evolution of the BBF structure in the magnetosphere. Double-check if your statements on lines 355-356 agree with the description on lines 149-151.

Text on lines 355–356 aims to explain how the BBF evolves from its initial generation onwards, while lines 149–151 describe the state of the BBF at specific simulation times  $t=500~\mathrm{s}$  and  $t=550~\mathrm{s}$  (Figures 1g and 1j). We believe that descriptions on both lines 355-356 and 149–151 refer to the same temporal evolution of a BBF generated by magnetic reconnection in Earth's magnetotail, but differ only in focus on simulation times and level of detail.

The text on lines 355-357 is modified as follows: "As this fast flow approaches Earth, its flow turns azimuthally duskward, while slowly expanding dawnward. This interaction induces flow vortices on the duskside and dawn side of the flow (see Supplementary Movie S1)".

10. Comment (10): Lines 366-367: Do you mean Figure 7b instead of Figure 7d? The enhanced ionospheric flow channel seems to correspond to the earthwardmost part of the BBF. Maybe specify that on these lines. Actually, the authors could discuss somewhere in section 3.x that one cannot see the enhanced ionospheric flows for the entire magnetospheric BBF channel. Could the authors say anything for the possible reason for that?

Yes, we meant Figure 7b, and the manuscript has been updated accordingly.

Regarding the reviewer's suggestion, it is correct that enhanced ionospheric flows are not seen across the entire BBF channel. Strong ionospheric signatures typically occur at the BBF flanks, where flow shear and braking generate FACs that couple effectively to the ionosphere. The central part of the BBF couples only weakly, and ionospheric conductivity further influences the visibility of these signatures.

To reflect this, the following text has been added in Section 3.4 of the manuscript: "It is important to note that an enhanced ionospheric flow channel is not observed along the entire magnetospheric BBF channel. Strong ionospheric signatures arise mainly at the flanks of the BBF, where flow shear generates FACs that couple efficiently into the ionosphere. In contrast, the central part of the BBF couples only weakly, and the visibility of the flows is further controlled by local ionospheric conductivity. As a result, the relatively broad BBF in the plasma sheet is typically associated with localised ionospheric flow channels rather than wide, uniform enhancements."

11. Comment (11)/Recommendation: Finally, I encourage the authors to continue to carry out studies related to BBFs and their ionospheric signatures using Vlasiator in the future,

for instance, when BBFs are observed in the different parts of the magnetotail, such as in the postmidnight region, for comparison. And using different simulation runs.

We thank the reviewer for this constructive suggestion. Indeed, future work will aim to examine the ionospheric signatures of dawnside BBFs using new Vlasiator simulation runs and compare these results with those from duskside BBFs.

**Title: Ionospheric signatures of a Bursty Bulk Flow in the 6D Vlasiator simulation**

Authors: Abiyot Bires Workayehu, +co-authors

**Replies to Referee #2**

We thank Referee #2 for comments and constructive suggestions on our manuscript. Below, we provide point-by-point responses to all comments. The referee's comments are presented in black, while our replies are shown in blue directly below each comment. We have carefully considered all suggestions and, where appropriate, made corresponding revisions in the manuscript. All changes in the manuscript are marked(added texts are shown in blue, and deleted texts are shown in red and struck through by a horizontal line). We hope our answers and the corresponding changes (where necessary) in the manuscript are satisfactory.

This paper presents numerical simulation results from the 6D hybrid-Vlasov code Vlasiator, coupled with an electrostatic ionospheric model, to investigate the ionospheric signatures of bursty bulk flows (BBFs). The study's core contribution lies in the detailed mapping of BBF-induced vortical flows to specific field-aligned current (FAC) systems and their associated ionospheric responses. These include enhancements in FACs, ionospheric conductances, and precipitating particle energies. The manuscript is well-written, and the results are clearly presented. The manuscript is clearly worth publishing, but before I could recommend publication, the authors could address a few minor comments, which are listed below.

1. Comment (1): The simulation uses steady and extreme solar wind conditions (Vsw = 750 km/s, Bz = -5 nT). How representative are these results of more typical solar wind conditions? Please justify this choice and, importantly, discuss how the results might change under more typical or variable solar wind conditions. For example, would a weaker IMF or lower solar wind velocity still produce such distinct ionospheric signatures?

We thank the reviewer for this comment. Yes, the solar wind driving conditions used in the Vlasiator simulation correspond to a fast solar wind stream (Vsw = 750 km/s) with moderately active conditions (Bz = -5 nT). The fast solar wind was chosen to speed up the initialisation phase of the simulation run. Importantly, this kind of simulation would be significantly more expensive computationally, and the present Vlasiator runs with these solar wind driving conditions are the best that can be performed with the current supercomputers.

To clarify our choice, we add the following text in Section 2.1: "The fast solar wind was chosen to speed up the initialisation phase of the simulation run (Palmroth et al., 2023), while Bz = -5 nT represents conditions favourable for magnetic reconnection without being strongly disturbed."

A stronger southward IMF and higher solar wind speed increase the coupling between the

solar wind and Earth's magnetosphere, leading to more frequent substorms and BBFs in the magnetotail (Zhang et al., 2016), and the same processes are expected during weaker conditions but with lower amplitude and lower occurrence rates (Zhang et al., 2016). Regarding ionospheric signatures, generally, stronger solar wind driving conditions result in more significant ionospheric effects and more pronounced signatures. A weaker solar wind driving may still produce less intense signatures, but not necessarily less distinct ones.

2. Comment (2): The BBF criterion ( $Vx \ge 400 \text{ km/s}$ ) is standard, but duration criteria are not discussed. How long does this BBF persist?

We thank the reviewer for this comment. This BBF persists for about 350 s, after which it interacts with other BBFs and evolves into a more complex plasma flow structure.

The following text is added in Section 3.1 of the manuscript: "The BBF persists for about 350 s, after which it interacts with other BBFs and evolves into a more complex plasma flow structure. This is consistent with observational studies, which typically report BBF durations ranging from a few minutes to 10 minutes (Baumjohann et al., 1990; Angelopoulos et al., 1992)."

3. Comment (3): The study focuses on a single BBF event within the simulation. Is it a typical or ideal BBF produced by the Vlasiator model? Have the authors observed similar signatures for other BBFs in their simulations?

The BBF analysed here is a typical BBF produced by the Vlasiator simulation. Yes, BBFs with more complex structures and comparable ionospheric signatures are also observed in the later stages of the simulation.

4. Comment (4): The duskside preference of BBFs is noted. How does this asymmetry influence the ionospheric signatures compared to dawnside BBFs?

This paper discusses signatures of a duskside BBF, and we have not observed an isolated dawnside BBF in the simulation to make a one-to-one comparison of ionospheric signatures. Therefore, it is difficult to draw conclusions about the signature of a dawnside BBF compared to the duskside. However, as shown in the present paper, the dusk preference of BBFs causes ionospheric signatures to cluster in the pre-midnight MLT sector, where we have observed enhanced FAC, conductance, and flow channel, while the post-midnight ionosphere displays weaker and spatially and temporally smooth distributions of these variables.

5. Comment (5): The simulation assumes a 0° dipole tilt, which is a simplification. Could including a more realistic dipole tilt affect ionospheric coupling?

Yes, including realistic dipole tilt angles can affect ionospheric coupling. Assuming a 0° tilt simplifies the geometry but neglects hemispheric and seasonal asymmetries that influence how the magnetosphere couples to the ionosphere. While using a 0° tilt angle roughly represents average equinoctial conditions, incorporating a realistic tilt changes the geometry of magnetopause reconnection, alters the mapping of field-aligned currents and convection into the ionosphere, and modifies ionospheric conductance through seasonal and diurnal variations in solar illumination.

6. Comment (6): The thin-shell approximation at 100 km altitude is justified, but how might altitude-dependent effects influence the results?

We thank the reviewer for this comment. The thin-shell approximation effectively captures large-scale ionospheric current systems and electric fields through height-integrated Pedersen and Hall conductances. However, we do not know the influence that altitude-dependent ionospheric profiles may have on the results. High-resolution volumetric measurements of ionospheric parameters, such as those anticipated from the upcoming EIS-CAT3D (McCrea et al., 2015), will help to resolve this.

7. Comment (7): A recent study by Kumar et al. (2025) (https://doi.org/10.1029/2024JA032953), using THEMIS and MMS observations, reported a dawn-dusk asymmetry in flows and a significant deceleration of these flows earthward of X <-15 RE. Please comment on whether the Vlasiator simulation reproduces a comparable braking effect in the near-Earth region.

The duskside preference and near-Earth deceleration of fast plasma flows observed in our study agree with Kumar et al. (2025), who observed significant flow braking earthward of X < -15 RE and more frequent occurrence in the premidnight region than postmidnight. However, the detailed braking effect reported by Kumar et al. is not explicitly addressed in this paper. Further study, directly comparing Vlasiator results with observations in this region, would be needed to assess whether the braking effect reported by Kumar et al. (2025) is reproduced.

- 8. Comment (8): On page 2, line 45, "breaking region" should likely be "braking region". Please check for this consistency throughout the manuscript. Yes, the reviewer is correct. The term is corrected to "braking region", and the manuscript has been carefully checked for consistency throughout.
- 9. Comment (9): Line 199-200: At t = 450 s, a BBF is active, yet the paper says there are "no significant ionospheric signatures." What is the expected delay between BBF arrival and ionospheric response in this setup?

Yes, at t = 450 s, a BBF is indeed present; however, it is too weak to generate significant vorticity, which in turn limits the strength of FACs and their manifestation as an ionospheric signature. The expected delay between the BBF and the ionospheric response at this simulation time is approximately 18s. This estimate is based on the Alfvén speed along the field line connecting the BBF location to the coupling radius at 5.6  $R_E$ , with an additional 2 s delay from the coupling radius to the ionosphere (Ganse et al., 2025) to obtain the total delay time.

The following text is added to Section 3.3 of the manuscript: "At t = 450 s, a BBF is indeed present; however, it is too weak to generate significant vorticity, which in turn limits the strength of FACs and their manifestation as an ionospheric signature."

10. Comment (10): The use of Bz = 0 as a proxy for reconnection lines is an oversimplification, as it does not inherently confirm the presence of active reconnection. Could the authors elaborate on how additional reconnection indicators—such as flow reversals, Hall magnetic field signatures, or localised energy conversion— align with the Bz = 0 regions in their simulation?

We thank the reviewer for highlighting the limitations of using  $B_z = 0$  as a proxy for reconnection lines. We fully agree that  $B_z = 0$  alone does not confirm active reconnection.

In our study,  $B_z = 0$  was used in Section 3.1 as a simplified indicator of potential X-line locations. To identify actual reconnection sites, we additionally applied the method by Alho et al. (2024), which distinguishes both X- and O-lines. This approach has been used in previous publications and provides a more robust and reliable identification of reconnection points in Vlasiator simulations. As shown in Figure 3 and Supplementary Movie S1, the X-lines obtained using this method reasonably align with the flow reversal at points where active reconnections occur.

Correspondence: Abiyot Workayehu (abiyot.workayehu@helsinki.fi)

[revised manuscript text omitted]

Using a combination of observational data and empirical magnetospheric models, Sergeev et al. (2020) studied investigated the origins and orientations of nightside auroral arcs. The authors found several stable auroral arcs associated with BBFs and indicated that the FACs associated with the observed arcs are generated by BBFs. Additionally, the authors discussed the orientation of auroral arcs in the ionosphere. They found that the majority of nightside arcs originate from the magnetotail current sheet region and argued that their orientation is influenced by the FACs associated with the BBFs and that the arcs tend to align with the direction of the electric field, 
[revised manuscript text omitted]